# Interference and co-existence of staphylococci and *Cutibacterium acnes* within the healthy human skin microbiome

Charlotte Marie Ahle [1,2✉], Kristian Stødkilde [3], Anja Poehlein [4], Mechthild Bömeke[4], Wolfgang R. Streit [2], Horst Wenck[1], Jörn Hendrik Reuter[1], Jennifer Hüpeden[1] & Holger Brüggemann [3✉]

Human skin is populated by trillions of microbes collectively called the skin microbiome. *Staphylococcus epidermidis* and *Cutibacterium acnes* are among the most abundant members of this ecosystem, with described roles in skin health and disease. However, knowledge regarding the health beneficial effects of these ubiquitous skin residents is still limited. Here, we profiled the staphylococcal and *C. acnes* landscape across four different skin sites of 30 individuals (120 skin samples) using amplicon-based next-generation sequencing. Relative abundance profiles obtained indicated the existence of phylotype-specific co-existence and exclusion scenarios. Co-culture experiments with 557 staphylococcal strains identified 30 strains exhibiting anti-*C. acnes* activities. Notably, staphylococcal strains were found to selectively exclude acne-associated *C. acnes* and co-exist with healthy skin-associated phylotypes, through regulation of the antimicrobial activity. Overall, these findings highlight the importance of skin-resident staphylococci and suggest that selective microbial interference is a contributor to healthy skin homeostasis.

[1] Beiersdorf AG, Research & Development, Front End Innovation, 20245 Hamburg, Germany. [2] Department of Microbiology and Biotechnology, University of Hamburg, 22609 Hamburg, Germany. [3] Department of Biomedicine, Aarhus University, 8000 Aarhus, Denmark. [4] Department of Genomic and Applied Microbiology, Institute of Microbiology and Genetics, University of Göttingen, 37073 Göttingen, Germany. ✉email: Charlotte.Marie.Ahle@studium.uni-hamburg.de; brueggemann@biomed.au.dk

Human skin is colonized by a diverse community of microorganisms, the composition of which is shaped by numerous host-related and external factors, including chemical and physical parameters, skin topography and microbe-microbe interactions[1].

*Staphylococcus* and *Cutibacterium* are known to be the most abundant and ubiquitous genera within the human skin microbiome[2,3], found across almost all parts of the skin ecosystem, albeit with preferential niches. Some species of staphylococci such as *S. epidermidis* are often located in sites of high humidity, while *C. acnes* is found more often in sebaceous areas[4–6]. Both genera are known to exhibit traits that have been linked to specific health- and disease-related states and are selectively regarded as key skin health sentinels[7–12].

*S. epidermidis* is the most abundant skin-colonizing coagulase-negative staphylococci (CoNS). The species is phylogenetically divided into three main clusters (A, B, C)[13–15] and is assigned to different sequence types (ST). Notably, *S. epidermidis* STs have been linked to nosocomial infections, suggesting relevance for pathogenic potential (e.g. ST2, ST5, ST23 and ST215)[16,17].

*C. acnes* is a polyphyletic species that can be divided into different subspecies and phylotypes, namely, IA$_1$, IA$_2$, IB, IC, II and III[18,19]. To enable characterisation of mixed populations of *C. acnes*, a single locus sequence typing (SLST) scheme has been developed that enables the differentiation into ten classes (A to L)[20]. SLST classes A to E correspond to phylotype IA$_1$ strains, whereas SLST classes F, G, H, K and L correspond to phylotypes IA$_2$, IC, IB, II and III, respectively[20]. Recent work has shown that some phylotypes/SLST classes are enriched in individuals with the skin disorder acne vulgaris, whereas others have been identified as markers of healthy skin. Acne-associated phylotypes include SLST classes A and C (both phylotype IA$_1$) and F (IA$_2$), whereas healthy skin is colonised with more diverse populations with a higher prevalence of strains belonging to the SLST classes H (IB) and K (II)[21–26].

A limited number of studies have indicated that *Staphylococcus* spp. and *Cutibacterium* spp. may be interacting in a strain-dependent manner. For instance, some staphylococcal strains can produce bacteriocins[27,28] or short-chain fatty acids[29], preventing the colonisation and spread of *C. acnes* and other disease-associated bacteria. However, there is still limited knowledge regarding interactions between the two most abundant genera on human skin.

Here, we used a combination of culture-dependent and -independent approaches to characterise staphylococcal and *C. acnes* populations within the healthy skin microbiome of 30 healthy individuals (four skin sites, 120 samples) and assess their potential for co-existence and mutual exclusion within this ecosystem.

An amplicon-based next-generation sequencing (NGS) method[20,30,31] was applied in tandem with in vitro antagonistic assays, whole genome sequencing of isolates and gene expression analysis to uncover selective exclusion and co-existence of acne- and healthy skin-associated *C. acnes* lineages, respectively, by staphylococcal strains. Our findings provide insights into the healthy skin microbiome landscape, revealing a key role of staphylococci in maintaining skin microbiome homoeostasis through microbial interference.

## Results

### Culture-dependent and -independent methods can determine staphylococcal populations with overall high congruency.
Samples for cultivation, amplicon-based NGS analysis and skin parameter measurements (hydration and sebum content) were taken from 30 healthy volunteers across four different skin sites (back, cheek, forearm and forehead; $n = 120$ samples) (Fig. 1a). 572 bacterial isolates were obtained via selective cultivation, of which 557 were identified as CoNS via MALDI-TOF mass spectrometry (Fig. 1b). Across all skin sites, the majority of isolates were identified as *S. epidermidis* ($n = 374$, 67.2%), followed by *Staphylococcus hominis* ($n = 86$, 15.4%). Forehead, cheek and back skin sites were dominated by strains of *S. epidermidis*, followed by *Staphylococcus capitis* (relative abundance of 74.5% and 11.7%, respectively), whereas on forearm skin sites, a larger number of strains of *S. hominis* and *Staphylococcus haemolyticus* (relative abundance of 38.7% and 7.3%, respectively) were isolated (Fig. 1b).

Moisture content (skin hydration) was highest on back and forehead skin as compared to cheek and forearm skin (Fig. 1c). The latter sites exhibited the highest sebum content and numbers of staphylococci per cm$^2$ (colony forming units/CFU), whereas forearm skin sites were particularly low in sebum and numbers of staphylococci (Fig. 1c).

Next, we applied an NGS approach based on the amplification of a specific section of the *tuf* gene[30] to molecularly characterise the resident staphylococcal populations on back, cheek, forearm and forehead skin samples. In 93 out of 120 samples, the *tuf* gene fragment could be amplified. In total, sixteen different staphylococcal species were identified in these 93 samples (Fig. 2a; Supplementary Data 1). The majority of skin sites were populated by several different co-existing staphylococcal species (on average 3.1 species) while in nearly one in ten samples (9.7%) only one staphylococcal species was identified. Across all skin sites tested, *S. epidermidis* was the most abundant species detected (average relative abundance 41.7%), followed by *S. capitis* (24.7%), *Staphylococcus saccharolyticus* (10.2%) and *S. hominis* (9.6%) (Fig. 2a, b).

The relative abundance profiles gained using the NGS-based amplicon approach were found to be in broad agreement with culture-based profiling (*S. epidermidis*, *S. hominis* and *S. capitis* were cultivated most frequently from samples). An exception was *S. saccharolyticus*, which was only detected using the NGS-based amplicon approach; this is likely due to the fastidious growth requirements of *S. saccharolyticus*[30]. *S. aureus* was only detected in cheek skin samples (relative abundance of 2.5%), possibly due to the proximity to the nasal cavity, the preferred niche of *S. aureus*[32] (Fig. 2a, b).

To test for differential abundance, we performed an Analysis of Compositions of Microbiomes with Bias Correction (ANCOM-BC) between all the four skin sites. The results showed that *S. hominis* was significantly more abundant in forearm skin samples compared to the other skin sites (Supplementary Table 1). Alpha diversity of staphylococcal populations was measured with the Shannon index and compared between the four skin sites. The highest staphylococcal diversity was observed in forearm skin samples, followed by back and cheek skin samples (Fig. 2c). Spearman correlation revealed a significant negative correlation of CFU count and staphylococcal alpha diversity (Fig. 2d). Spearman correlation analysis between staphylococcal species abundance and skin parameters showed that the abundance of *S. hominis* correlated with staphylococcal alpha diversity, and inversely correlated with CFU count and sebum content (Fig. 2e). The correlation analysis was performed for each skin site separately; on back, cheek and forehead skin the positive correlation between *S. hominis* and staphylococcal alpha diversity was observed (Supplementary Fig. 1).

### S. epidermidis strains from healthy human skin are highly diverse and belong to non-nosocomial-associated phylogenetic lineages.
Given the abundance of *S. epidermidis* across all skin

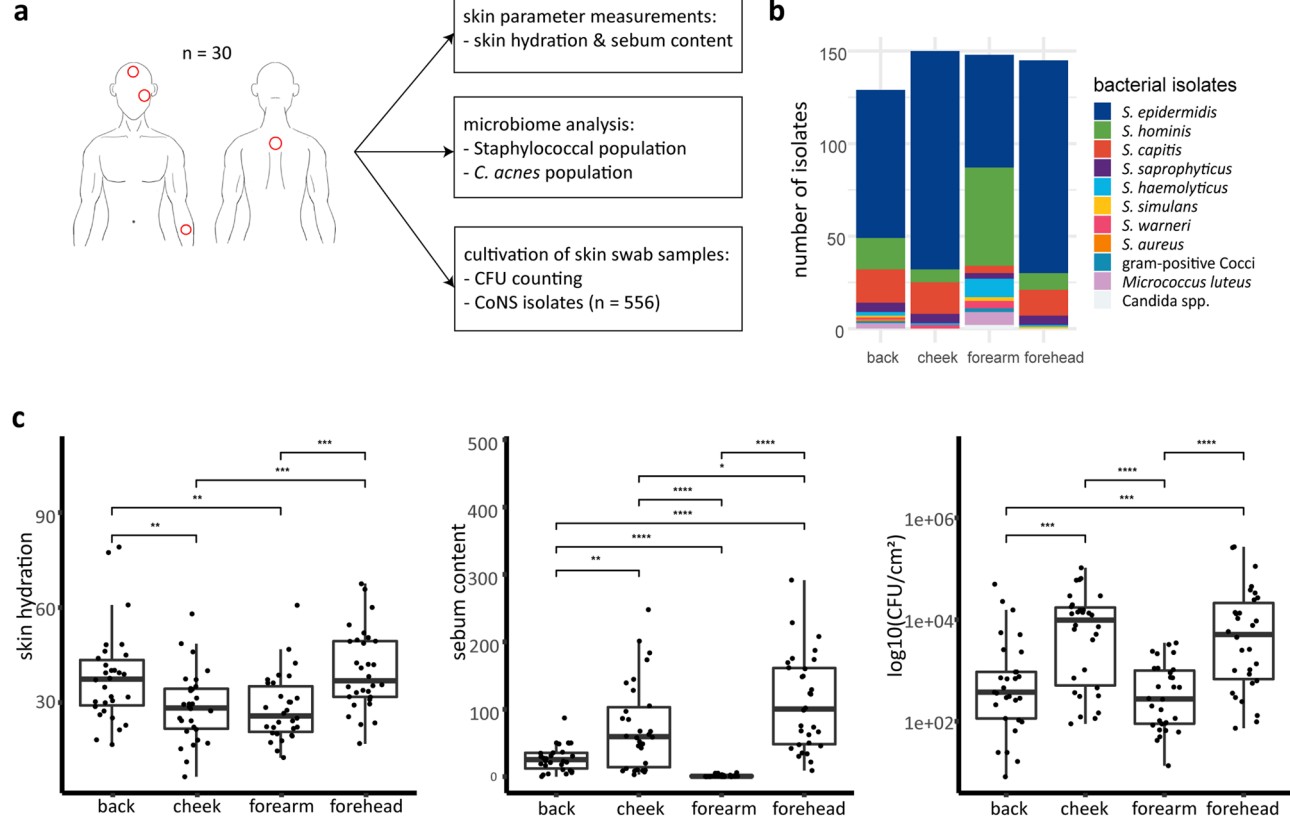

**Fig. 1 Skin parameters and distribution of staphylococcal isolates on four different skin sites. a** Study design. **b** Number of isolates of identified bacterial species per skin site. A selective cultivation approach to primarily isolate staphylococci was applied. **c** Skin hydration, sebum content and CFU per cm² on back, cheek, forearm, and forehead skin ($n = 30$ for each skin site. $*p \leq 0.05$, $**p \leq 0.01$, $***p \leq 0.001$, $****p \leq 0.0001$. Unpaired Wilcoxon test). Middle lines of boxplots indicate the median. Lower and upper lines represent the first and third quartiles. Whiskers show the 1.5x inter-quartile ranges.

sites, we wanted to further delineate the population structure of the cultured isolates derived from healthy skin. 69 isolates were genome-sequenced and phylogenetically compared to 286 previously published *S. epidermidis* genomes (Fig. 3a). The sequenced *S. epidermidis* strains were highly diverse with distinct strain individuality, as judged by analysis of the pan-genome (Supplementary Fig. 2). We also found that the accessory genome of *S. epidermidis* is substantially larger than that of *C. acnes* (Supplementary Fig. 3).

To further delineate the *S. epidermidis* strain diversity, the genomes were assigned to the three clades[13–15], with 42 strains clustering to clade A, four to clade B and 23 to clade C (Fig. 3a). Only four isolates clustered within clade B, a clade that is thought to exhibit reduced pathogenic potential as compared to clades A and C[14]. Interestingly, a high number of isolates were assigned to clade C, which has not previously been associated with staphylococcal isolates from healthy skin.

Among this 69-strain cohort, strains belonging to 23 different STs and additional eight so far not described STs were found (Supplementary Data 2). Most common were isolates of ST19 (nine isolates from five people), ST73 (nine isolates from seven people) and ST65 (eight isolates from six people). From 19 individuals we have isolated and sequenced more than one *S. epidermidis* isolate (two to six isolates per person). Almost all individuals (18/19) carried multiple strains belonging to different STs (Supplementary Data 2). Notably, none of the *S. epidermidis* strains isolated here belonged to known infection-associated STs, namely the described types ST2, ST5, ST23 and ST215[16,17].

To further ascertain strain-specific traits, we checked for the presence of *mecA, icaA* and IS256, genes known to be more prevalent in infection-associated *S. epidermidis* compared to commensal isolates[13,33]. Out of the 69 *S. epidermidis* genomes sequenced, the *mecA* gene was identified in four and *icaA* in 18. Only one strain was found to have both *mecA* and *icaA*, and IS256 was not identified in any of the genomes analysed (Fig. 3b).

**Application of an amplicon-based NGS method enables profiling of resident *C. acnes* populations at phylotype resolution.** To gain deeper insight into the landscape of *C. acnes* populations resident on healthy human skin, we next applied a previously developed SLST amplicon-based NGS scheme to the 120 samples[20]. In 113 out of 120 samples, the SLST fragment could be amplified. In total, 39 different *C. acnes* SLST types were identified in these 113 samples (Supplementary Data 3). All of the ten *C. acnes* SLST classes (A to L) were found across the different samples from cheek and forearm skin, whereas the B- and G-class *C. acnes* were absent from forehead and back skin, respectively (Fig. 4a; Supplementary Data 3). On average, 3.6 different *C. acnes* SLST classes were found resident at each skin site.

Across all skin sites, *C. acnes* strains belonging to the IA₁ phylotype were most frequently detected (average 58.1%) (Fig. 4a). Among the IA₁ phylotype, SLST class A was the most abundant (27.6%), followed by SLST classes D (20.7%) and C (5.9%). The second most abundant *C. acnes* phylotype was II (corresponding to SLST class K) (19.2%), followed by IB (corresponding to SLST class H) (12.2%) (Fig. 4a, b).

A-class *C. acnes* had a similar average relative abundance across all four skin sites (24.7–30.0%). A-class *C. acnes* were most abundant on cheek, forearm and forehead skin, whereas on back

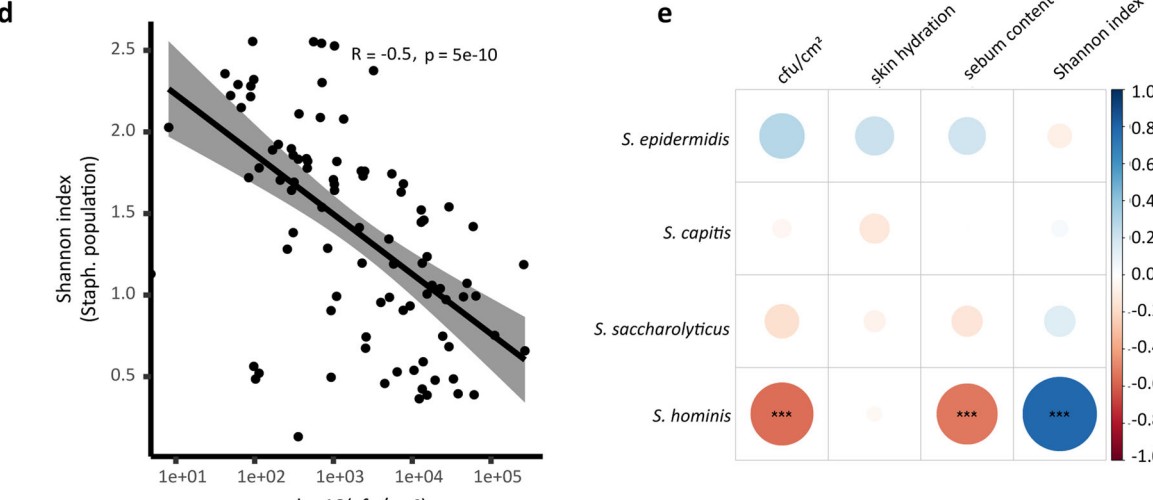

skin D-class *C. acnes* were dominant (37.3%) (Fig. 4b). An ANCOM-BC analysis confirmed this observation and showed a significant higher abundance of D-class *C. acnes* in back samples compared to forearm and forehead samples (Supplementary Table 2). As expected, cheek and forehead skin had a more similar *C. acnes* SLST class composition, compared to back and forearm skin (Fig. 4b). L-class *C. acnes* were more abundant on the forearm skin (average of 5.0%) compared to the other three skin sites (<0.7%).

Similar to the observation regarding staphylococcal populations (Fig. 2c), the alpha diversity of *C. acnes* populations from forearm skin samples was higher compared to the other three skin sites (Fig. 4c). Regarding the influence of skin parameters on *C. acnes* populations, it was found that the alpha diversity

**Fig. 2 Staphylococcal populations in skin samples determined by amplicon-based NGS and correlation to skin parameters. a** 93 out of 120 samples, obtained from 30 volunteers, were positive for the tuf2 amplification; relative abundance of staphylococcal species on back, cheek, forearm and forehead skin are shown (see also Supplementary Data 1). **b** Stacked bar plot showing mean values of relative abundances of staphylococcal species overall and for the four skin sites. **c** Shannon diversity index of staphylococcal population per skin site showed highest diversity in forearm skin samples (back skin samples, $n = 20$; cheek skin samples, $n = 26$; forearm skin samples, $n = 21$; forehead skin samples, $n = 26$. **$p \leq 0.01$, ***$p \leq 0.001$, ****$p \leq 0.0001$. Unpaired Wilcoxon test). Middle lines of boxplots indicate the median. Lower and upper lines represent the first and third quartiles. Whiskers show the 1.5x inter-quartile ranges. **d** Spearman correlation between Shannon index of staphylococcal populations and CFU per $cm^2$. **e** Spearman correlation between staphylococcal species abundance and skin parameters. The colour code illustrates the correlation coefficient; blue colour represents positive correlation (0 to 1) and red colour inverse correlation (−1 to 0). For instance, the presence of *S. hominis* strongly (and statistically significant) correlated with the Shannon index and inversely correlated with CFU count and with sebum content (FDR-adjusted *p*-value, ***$p \leq 0.001$).

positively correlated with the abundance of K-class *C. acnes* (Fig. 4d).

**Staphylococcal isolates exhibit antimicrobial activity against acne- but not healthy skin- associated *C. acnes* phylotypes.** Correlation of the relative abundances of the four most abundant staphylococcal species and *C. acnes* SLST classes revealed a significant positive correlation between *S. epidermidis* and K-class *C. acnes*, as well as an inverse correlation between *S. epidermidis* and A-class *C. acnes*, albeit statistically non-significant (Fig. 5a). We therefore wanted to further understand the potential for microbial interference between staphylococcal and *C. acnes* populations.

First, we conducted in vitro antagonistic assays to ascertain the antimicrobial activity of our isolated CoNS strains. All 572 isolates were screened against a *S. aureus* strain and an A-class *C. acnes* strain (Supplementary Table 3) to identify staphylococcal isolates with bioactivity. The 30 strains identified with activity against the *C. acnes* strain were then further screened against eleven different *C. acnes* strains covering six different SLST classes (A, C, D, H, K, L) (Table 1), including both acne (SLST classes A and C)- and healthy skin-associated types (SLST classes H and K), in order to identify any phylotype-specific bioactivity. In total, 4% (22/557) of the tested staphylococcal isolates exhibited activity against *S. aureus* and 5% (30/557) of them against one or more *C. acnes* strains. Of these 30 staphylococcal isolates, 17 were identified as *S. capitis*, six as *S. hominis*, five as *S. epidermidis* and two as *S. warneri*. Strains belonging to different *C. acnes* phylogenetic clades showed a remarkably different susceptibility to the antimicrobial activity of the various staphylococcal isolates. The two A-class *C. acnes* strains, DSM1897 and 12.1.L1, were most susceptible to this bioactivity, being inhibited by the antimicrobial activity and were inhibited by a total of 29 and 15 staphylococcal strains, respectively. In contrast, only one staphylococcal strain (*S. epidermidis* HAC26) was found to exhibit inhibitory bioactivity against D-class and H-class *C. acnes* strains (Table 1).

To provide insight into the in vivo relevance of these observations, we compared the relative abundance profiles of *C. acnes* populations originating from skin samples with and without the presence of antimicrobial active staphylococcal strains. Notably, the relative abundance of A-class *C. acnes* was significantly lower in skin sites from which a staphylococcal strain was isolated that exhibited antimicrobial activity in the antagonistic assay (Fig. 5b, c, Table 1). This inverse correlation of abundance was most pronounced in back skin samples; in samples containing antimicrobial-active staphylococci, there was a marked increase in the relative abundance of D-class *C. acnes* and a corresponding decrease in A-class *C. acnes*, respectively ($p = 0.004$ and $p = 0.0013$, respectively) (Supplementary Fig. 4).

**Selective regulation of the antimicrobial activity of *S. epidermidis* in response to sensitive and tolerant *C. acnes* strains.** In an initial effort to gain insight into the mechanisms underlying

the selective bioactivity of CoNS strains in response to different *C. acnes* strains (Fig. 5), we undertook genome-wide transcriptional analyses in co-culture experiments.

As a model organism, we chose *S. epidermidis* HAF242, as it was shown to exhibit antimicrobial activity against the A-class *C. acnes* strain DSM1897, but no lethal effects on the D-class *C. acnes* strain 30.2.L1 (Table 1). Genome sequencing of *S. epidermidis* HAF242 revealed the presence of the epidermin biosynthesis cluster (locus tag: LZT96_12010), which has previously been described as an important antimicrobial determinant[34]. For co-culture experiments, *S. epidermidis* HAF242 was inoculated on a lawn of *C. acnes* DSM1897 and 30.2.L1, respectively. As expected, colonies of *S. epidermidis* HAF242 showed clear inhibition zones on the lawn of *C. acnes* DSM1897, but not on the lawn of *C. acnes* 30.2.L1 (Fig. 6a). Attempts to reproduce this observation in liquid culture failed, indicating that solid surface growth and/or direct contact is a requirement for antimicrobial activity. Plate-grown co-cultures were harvested, subjected to RNA-sequencing and transcriptome analysis was carried out to determine any differential expression of *S. epidermidis* HAF242 genes in the two different co-cultures, representing sensitive and tolerant scenarios, respectively (*S. epidermidis* HAF242/*C. acnes* 30.2.L1 versus *S. epidermidis* HAF242/*C. acnes* DSM1897). Comparison of the transcriptome profiles from the co-culture experiments, revealed 33 significantly differentially expressed *S. epidermidis* genes: six genes were down-regulated, while 27 genes were up-regulated (Fig. 6b, Supplementary Data 4). Notably, when in co-culture with *C. acnes* 30.2.L1, *S. epidermidis* HAF242 exhibited a three-fold down-regulation of the quorum-sensing auto-inducing peptide gene (*agrD*) (in comparison to co-culture with *C. acnes* DSM1897). Furthermore, three phenol-soluble modulin (PSM) beta genes (*psmβ1, psmβ2, psmβ3*) and the precursor peptide gene for epidermin (*epiA*) were also significantly downregulated (Fig. 6b, Supplementary Data 4). To determine whether the differential gene expression profiles observed were due to up-regulation in response to *C. acnes* DSM1897 (sensitive strain) or down-regulation in response to *C. acnes* 30.2.L1 (tolerant strain), we conducted a comparative transcriptome analysis of *S. epidermidis* in monoculture (hereafter labelled as "control") (Supplementary Data 5 and 6). The resulting analysis revealed that the three *psm* genes were downregulated 3- to 11-fold in *S. epidermidis* grown in co-culture with *C. acnes* 30.2.L1 (tolerant strain) when compared to the control (Fig. 6c, Supplementary Data 6). In addition, the *agrD* gene was mildly down-regulated (2-fold), albeit not statistically significant. This indicates that *C. acnes* 30.2.L1, in contrast to *C. acnes* DSM1897, exerts an inhibitory effect on the antimicrobial activity of *S. epidermidis* HAF242 by down-regulating the expression of genes involved in anti-*C. acnes* activity.

**Discussion**

Here we highlight the importance of skin-resident staphylococci and the potential role of selective microbial interference for

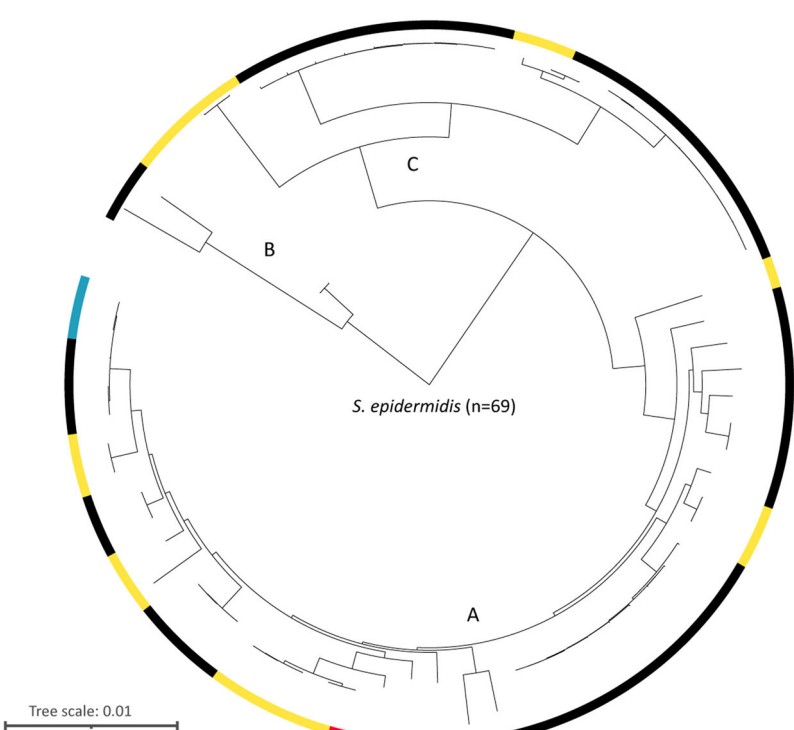

**Fig. 3 Phylogeny of *S. epidermidis* strains obtained in this study.** Phylogenetic trees are based on single nucleotide variants (SNVs) of the core genomes. **a** 286 *S. epidermidis* strains (genomes taken from RefSeq) (= black) and 69 strains isolated in this study (= green). Nosocomial sequence types (ST2, ST5, ST23) (= red) and non-nosocomial sequence types (= grey) are depicted. **b** 69 *S. epidermidis* strains isolated in this study. Highlighted are strains with *mecA* gene (= blue), *icaA* gene (= yellow) and one strain with *mecA* + *icaA* gene (= red).

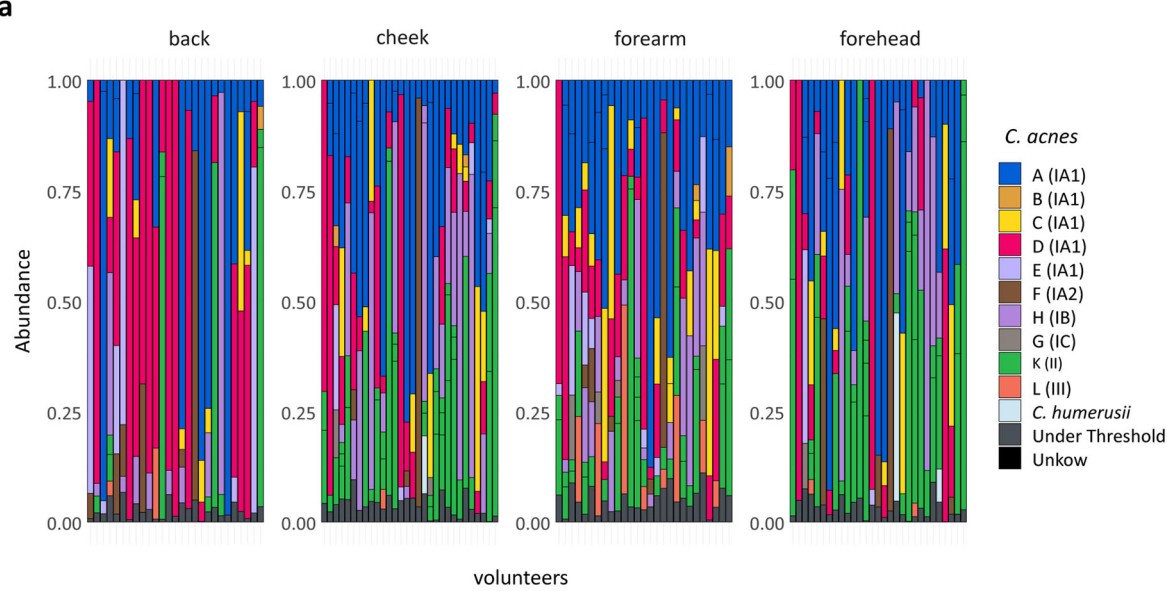

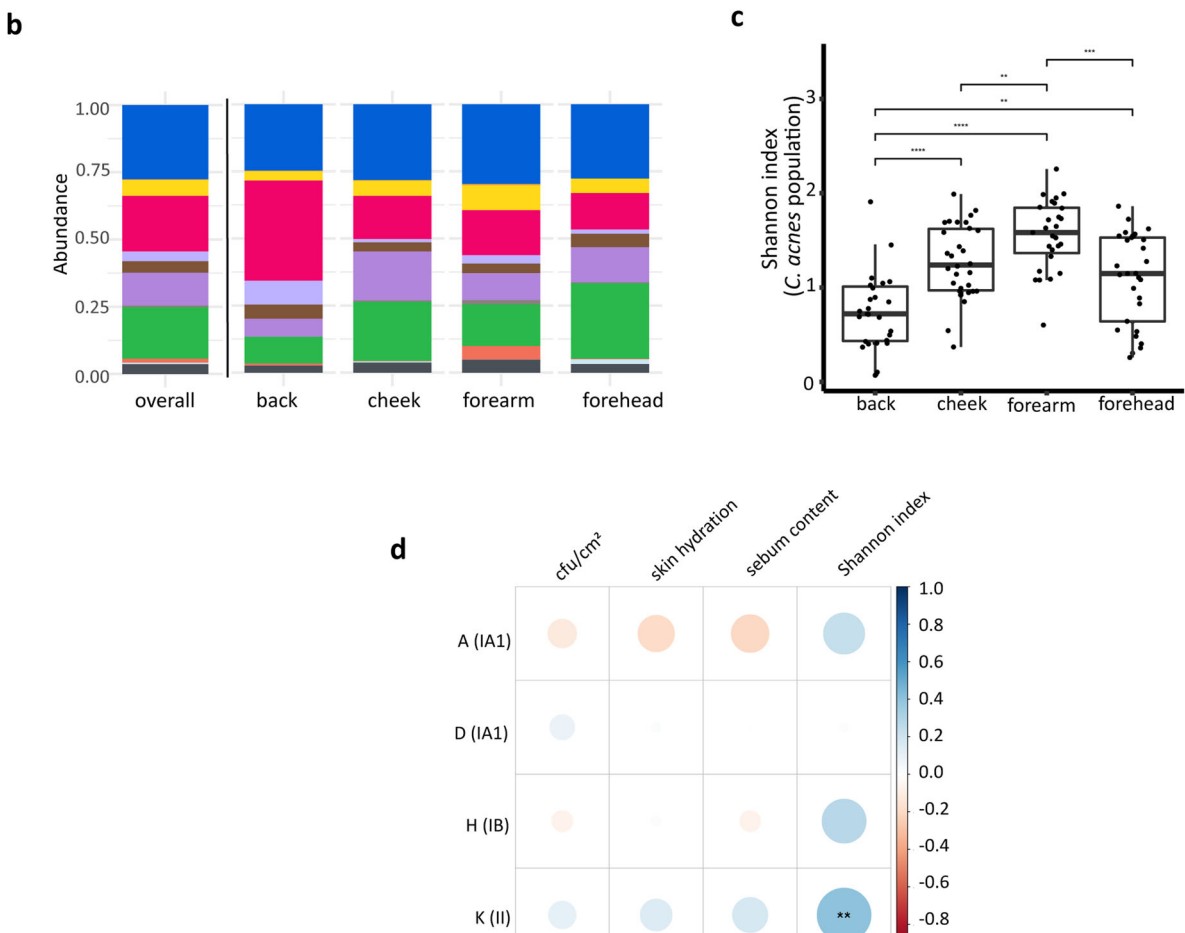

healthy skin homoeostasis. Our analysis, encompassing both culture-dependent and -independent methodologies, reveals diverse populations of staphylococci resident on healthy skin with selective microbial interference activity. The tandem application of the amplicon-based NGS methods used within this study enabled a detailed delineation of the diversity of both staphylococci and *C. acnes* populations resident across multiple skin sites

to provide species and phylotype level resolution, respectively. The resident staphylococci were found to exhibit phylotype-specific antimicrobial activity against acne-associated *C. acnes*, while co-existing with *C. acnes* phylotypes that are more commonly associated with healthy skin.

In agreement with previous studies[2,3,35], *S. epidermidis* was found to be the most abundant staphylococcal species detected

**Fig. 4 *C. acnes* populations in skin samples determined by amplicon-based NGS and correlation to skin parameters. a** 113 out of 120 samples, obtained from 30 volunteers, were positive for the SLST fragment amplification; relative abundances of *C. acnes* SLST classes of back, cheek, forearm and forehead skin are shown (see also Supplementary Data 3). **b** Stacked bar plot showing mean values of relative abundances of *C. acnes* SLST classes overall and for the four skin sites (for colour code see a). **c** Shannon diversity index of *C. acnes* populations per skin site (back skin samples, $n = 27$; cheek skin samples, $n = 30$; forearm skin samples, $n = 27$; forehead skin samples $n = 29$. **$p \le 0.01$, ***$p \le 0.001$, ****$p \le 0.0001$. Unpaired Wilcoxon test). Middle lines of boxplots indicate the median. Lower and upper lines represent the first and third quartiles. Whiskers show the 1.5x inter-quartile ranges. **d** Spearman correlation between relative abundances of *C. acnes* SLST classes and skin parameters. The colour code illustrates the correlation coefficient; blue colour represents positive correlation (0 to 1) and red colour inverse correlation ($-1$ to 0). For instance, the presence of *C. acnes* class K correlated (statistically significant) with the Shannon index. The correlation analysis is only shown for the four most frequently detected *C. acnes* SLST classes; correlation analysis for less frequent SLST classes that are detected in relatively few samples is not reliable (FDR-adjusted *p*-value, **$p \le 0.01$).

across all skin sites tested, followed by *S. capitis* and *S. saccharolyticus*. *S. epidermidis* is known both as a skin commensal and an opportunistic pathogen, the latter especially in infections of indwelling devices[36]. Delineation of the *S. epidermidis* strains into clades and STs within the current study revealed that clonal lineages often associated with an elevated pathogenic potential were rarely found on healthy skin: None of the 69 *S. epidermidis* isolates from healthy skin belonged to prominent infection-associated sequence types, such as ST2, ST5, ST23 and ST215[16,17] (clade A). However, only four of the 69 *S. epidermidis* skin isolates were classified here as belonging to the B-clade, which is thought to consist mainly of commensal skin isolates[13,14]. In addition, many strains were classified as belonging to the C-clade, for which very little knowledge is currently available[14]. The classification of the isolated strains across the range of different clades indicates that the current assignment provides limited information in terms of a strain's particular health-beneficial or -detrimental properties. This assumption supports previous studies that have highlighted the difficulty of using core-genome-derived phylogeny to differentiate pathogenic and commensal *S. epidermidis* strains[13,33], due to the fact that pathogenic traits can be acquired through horizontal gene transfer[15]. Sequences associated with pathogenicity such as the methicillin-resistance gene *mecA*, the biofilm operon *icaADBC* and the insertion sequence element IS256 are part of the accessory genome, and can thus be present in phylogenetically distinct strains[13,33]. Of the 69 *S. epidermidis* strains sequenced here, only 5.8% were positive for *mecA*, 26.1% for *icaADBC* and none for IS256, aligning with previous studies of commensal *S. epidermidis* (*icaA*: 13.3% and 33.8%; IS256: 0% and 4.2%; *mecA*: 6.7% and 15.5%)[13,33].

In contrast to *S. epidermidis*, with its open-pangenome and variable genome content, *C. acnes* is more conserved with a relatively limited accessory genome[5] (Supplementary Figs. 2 and 3). Core genome-based phylogeny divides *C. acnes* populations into six main phylotypes. In total, 178 different SLST types (medbac.dk/slst/pacnes; status: 15th of January 2022) belonging to ten SLST classes (A-L) have been reported[20]. Here, we identified 39 distinct SLST types that covered all ten SLST classes, within the 120 healthy skin samples profiled. Overall, highest relative abundances were determined for A-class *C. acnes* (27.6%), followed by D-class (20.7%), K-class (19.2%) and H-class (12.2%) *C. acnes*. All other SLST classes had lower average abundances, ranging between 5.9% and 0.2%.

As yet, the SLST amplicon-based NGS method has not been used for samples from diseased skin, such as acne vulgaris-affected skin. Thus, a direct comparison of our data with samples from diseased skin is not possible at present. However, the SLST scheme has been used in culture-dependent studies, albeit with low patient numbers, highlighting that A-, C- and F-class *C. acnes* strains are primarily associated with acne vulgaris[21,25]. In addition, previous studies using other schemes for determining the phylogenetic basis of *C. acnes* isolates, such as MLST, have found similar results[18,24,37]. These studies have also revealed that

healthy skin-associated strains often belong to the SLST classes H and K. In our study, H- and K-class *C. acnes* were also found at high relative abundance. Recent studies and reviews have highlighted possible genetic and physiological differences between acne-affected skin- and healthy skin-associated strains[7,38,39]. One important difference might be related to porphyrin; it was found that porphyrin production is more increased in acne-associated *C. acnes* strains, resulting in enhanced inflammasome activation in exposed keratinocytes[40,41]. However, the delineation of acne- and healthy skin-associated SLST classes might be oversimplified, as it seems likely that a high diversity of strains belonging to different SLST classes forms the basis of a healthy skin microbiome and thus, the loss of diversity is associated with acne[42]. We also noted a high relative abundance of D-class *C. acnes*, especially on back skin samples. There is currently very limited information available on this SLST class. Given the dominance of this lineage on back skin sites of multiple healthy individuals, we propose that it could also be health-associated.

The in vitro antagonistic assays conducted here enabled us to identify a range of CoNS strains with antimicrobial activity against A-class *C. acnes*, a class that is overrepresented in acne-affected skin[21–24]. This observation of phylotype-specific activity against disease-associated *C. acnes* strains is in contrast to previous work[27,28]. Our data indicate that active staphylococcal strains have the potential to modify the composition of resident *C. acnes* populations on skin. Importantly, we noted an inverse relationship between the abundance of staphylococcal strains exhibiting antimicrobial activity and specific phylotypes of *C. acnes*. Skin sites with resident antimicrobial-active staphylococci had a significantly lower abundance of A-class *C. acnes* in comparison to those lacking these active CoNS strains. These observations align with previous work that reported a decreased abundance of staphylococcal strains with antimicrobial activity on atopic dermatitis-affected skin compared to healthy controls[43]. Moreover, a low abundance of competitive staphylococcal strains was found to correlate with *S. aureus* colonization[43], a species that is often found on atopic dermatitis lesions[44] and is associated with disease severity[45].

To gain insight into the mechanisms underlying the phylotype-specific microbial interference observed, we conducted co-culture experiments using the antimicrobial-active *S. epidermidis* strain HAF242, and a tolerant D-class strain (30.2.L1) or a sensitive A-class *C. acnes* strain (DSM1897) and analysed the resulting transcriptome profiles. Here, we observed differential expression of genes encoding the lantibiotic epidermin precursor peptide EpiA and the phenol soluble modulin PSMβ. The activity of epidermin, a large peptide antibiotic, against *C. acnes* has been previously reported[46], but the role of PSMβ is still not understood. Interestingly, a *S. capitis* strain was recently identified that secretes four PSMs, which act synergistically as antimicrobials against *C. acnes*[27], opening up the possibility that PSMβs might contribute together with epidermin to the antimicrobial activity of *S. epidermidis* HAF242.

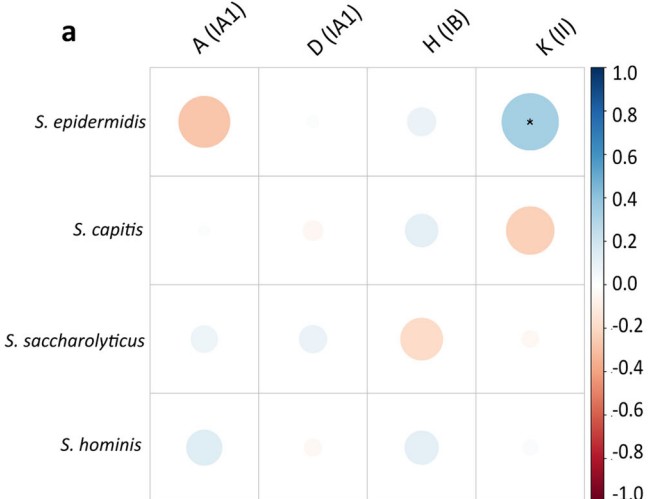

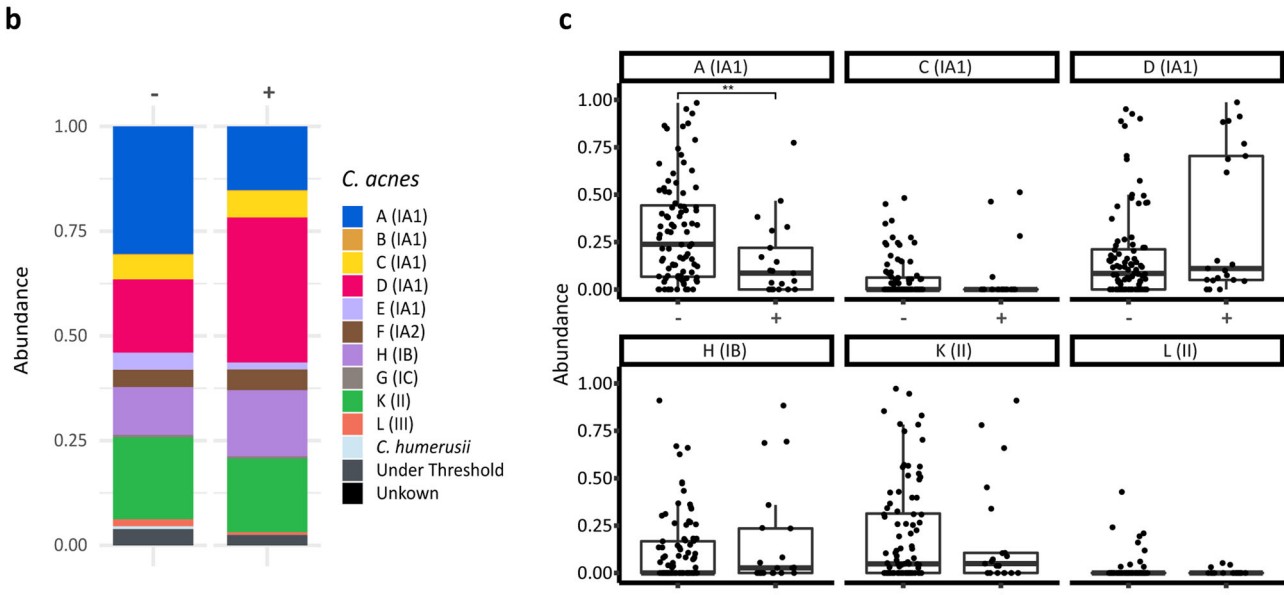

**Fig. 5 Staphylococci and *C. acnes* co-existence and inhibition profiles. a** A Spearman correlation between the four most abundant *Staphylococcus* species and *C. acnes* SLST classes found on the skin was performed. The colour code illustrates the correlation coefficient; blue colour represents positive correlation (0 to 1) and red colour inverse correlation (−1 to 0). For instance, this revealed a positive (and statistically significant) correlation between *S. epidermidis* and K-class *C. acnes* and an inverse correlation (albeit not statistically significant) between *S. epidermidis* and A-class *C. acnes*. The correlation analysis is only shown for the four most frequently detected staphylococci and *C. acnes* SLST classes; correlation analysis for less frequent staphylococcal species and *C. acnes* SLST classes that are detected in relatively few samples is not reliable. (FDR-adjusted *p*-value; *$p \leq 0.05$). **b** The mean relative abundances of *C. acnes* SLST classes on skin sites with (+) and without (−) antimicrobially active staphylococcal strains are depicted. The presence of staphylococcal strains with antimicrobial activity led to a decrease of the relative abundance of A-class *C. acnes*. **c** Boxplots of relative abundances of the six *C. acnes* SLST classes on skin sites with (+) and without (−) antimicrobially active staphylococcal strains ($n = 21$ (+) and $n = 92$ (−), respectively) are shown (FDR-adjusted *p*-value, **$p \leq 0.01$. Unpaired Wilcoxon test). Middle lines of boxplots indicate the median. Lower and upper lines represent the first and third quartiles. Whiskers show the 1.5x inter-quartile ranges.

The transcriptome analyses also highlighted a potential role for *agrD*, which is part of the *agr* quorum sensing (QS) system that encodes an autoinducing peptide (AIP) belonging to type I AIPs. It is detected by the histidine kinase AgrC, which in turn activates the response regulator AgrA. AgrA directly binds to the promotor region of target genes such as the *psm* locus and activates their expression[47,48]. In our experiments, co-culture with the D-class *C. acnes* strain led to down-regulation of the expression of *agrD*, *epiA* and *psm*β in *S. epidermidis*. Thus, we hypothesize that

D-class *C. acnes* can interfere with the QS system of *S. epidermidis* and suppresses the production and activity of antimicrobial peptides (Supplementary Fig. 5). This interference with the *agr* QS system has been observed previously: inter- and intraspecies interference of staphylococci through their *agr* QS system ("quorum quenching") can alter the expression of various target genes related to virulence and biofilm formation[6,49–51]. While these interactions occur mainly between staphylococcal species[6,49–51], one study found that *Candida albicans* can

**Table 1 Antimicrobial activity of staphylococci against *C. acnes* strains from six different SLST classes and *S. aureus* DSM799.**

| Indicator strains / CoNS strain | *C. acnes* class[a] A[b] | | C[b] | D | | H | | | K | | L | *S. aureus* DSM799 |
|---|---|---|---|---|---|---|---|---|---|---|---|---|
| *S. capitis* HAB177 | + | + | − | − | − | − | − | − | − | − | − | − |
| *S. capitis* HAB198 | + | + | − | − | − | − | − | − | − | − | − | − |
| *S. capitis* HAB200 | + | + | − | − | − | − | − | − | − | − | − | − |
| *S. capitis* HAB276 | + | + | − | − | − | − | − | − | − | − | − | − |
| *S. capitis* HAB277 | + | + | − | − | − | − | − | − | − | − | − | − |
| *S. capitis* HAB278 | + | + | − | − | − | − | − | − | − | − | − | − |
| *S. capitis* HAB280 | + | + | − | − | − | − | − | − | − | − | − | − |
| *S. capitis* HAB56 | + | + | − | − | − | − | − | − | − | − | − | − |
| *S. capitis* HAC349 | + | − | − | − | − | − | − | − | − | − | + | − |
| *S. capitis* HAC470 | + | − | − | − | − | − | − | − | − | − | − | − |
| *S. capitis* HAC49 | + | + | − | − | − | − | − | − | − | − | − | − |
| *S. capitis* HAC507 | + | − | − | − | − | − | − | − | − | − | − | − |
| *S. capitis* HAC508 | + | − | − | − | − | − | − | − | − | − | − | − |
| *S. capitis* HAC509 | + | − | − | − | − | − | − | − | − | − | − | − |
| *S. capitis* HAC510 | + | − | − | − | − | − | − | − | − | − | − | − |
| *S. capitis* HAF401 | + | − | − | − | − | − | − | − | − | − | + | − |
| *S. capitis* HAF403 | + | − | − | − | − | − | − | − | − | − | + | − |
| *S. epidermidis* HAC26 | + | + | + | + | + | − | + | − | − | − | + | + |
| *S. epidermidis* HAC588 | + | − | − | − | − | − | − | − | + | − | + | − |
| *S. epidermidis* HAC590 | + | − | − | − | − | − | − | − | + | − | + | − |
| *S. epidermidis* HAF242 | + | − | − | − | − | − | − | − | + | − | + | − |
| *S. epidermidis* HAF424 | + | − | − | − | − | − | − | − | + | − | + | − |
| *S. hominis* HAA254 | + | − | − | − | − | − | − | − | + | + | − | − |
| *S. hominis* HAA272 | + | + | + | − | − | − | − | − | + | + | + | − |
| *S. hominis* HAA273 | + | + | + | − | − | − | − | − | + | + | + | − |
| *S. hominis* HAA274 | + | + | + | − | − | − | − | − | + | + | + | − |
| *S. hominis* HAB257 | + | − | − | − | − | − | − | − | + | + | − | − |
| *S. hominis* HAC286 | − | − | − | − | − | − | − | − | − | − | + | + |
| *S. warneri* HAA333 | + | + | + | − | − | − | − | − | + | + | + | + |
| *S. warneri* HAA334 | + | + | + | − | − | − | − | − | + | + | + | + |

[a]The following *C. acnes* strains were used: A class, DSM1897 and 12.1.L1; C class, 15.1.R1; D class, 30.2.L1 and 09-193; H class, 11-90, KPA171202 and 21.1.L1; K class, 11-49 and 11-79; L class, PMH5.
[b]These SLST classes are enriched on acne-affected skin.

interfere with the alpha toxin production of *S. aureus* via the *agr* system[52]. It needs to be proven in future studies if *C. acnes*, in a phylotype-specific manner, can interfere with the *agr* QS system of staphylococci. This interference might not only have a benefit for *C. acnes*, i.e. guaranteeing its survival, but also for the staphylococcal strain exhibiting antimicrobial activity. It was shown that the production of antimicrobial peptides negatively affects the growth rate of the producing *S. epidermidis* strain[53]. Therefore, the suppression of antimicrobial peptide production could also be beneficial for the staphylococcal strain.

The insights gained within this study are of course framed within the confines of relatively small sample size (30 individuals) and the semi-quantitative nature of the data generated by the amplicon-based NGS methods applied here (relative abundance). We also use two distinct NGS methods for profiling the CoNS and *C. acnes* populations, with differing analytical scope: The SLST scheme used for *C. acnes* populations provides phylotype resolution, whereas the tuf2 scheme used to dissect CoNS populations offers species level identification. Further work is required to dissect the CoNS population with ST- or strain-level resolution, possibly, by using separate yet to be established SLST approaches for each CoNS species; one approach was recently developed for *S. epidermidis*[54]. Furthermore, only a randomly selected subset of *S. epidermidis* isolates was genome-sequenced and STs were assigned. Thus, it cannot be completely ruled out that some of the volunteers carry *S. epidermidis* isolates that belong to infection-associated STs.

Overall, however, our results provide further insight into the importance of commensal staphylococci on healthy human skin and their crucial role for *C. acnes* population homoeostasis. The knowledge and insights gained regarding the potential of CoNS strains to exclude and co-exist with disease- and healthy skin-associated *C. acnes* phylotypes has potential relevance for skin health maintenance and customized bacteriotherapy; for instance, applied to skin disorders that are associated with dysbiosis of *C. acnes* populations such as acne vulgaris.

## Methods

**Cohort and sample acquisition**. Swab samples were collected from 30 volunteers (female, $n = 14$; male, $n = 16$) with an age range of 22–43 years from forehead, cheek, back and forearm skin, as described previously[30]. In brief, an area of 25 cm$^2$ of forehead, cheek, back skin and 50 cm$^2$ on forearm skin was swiped with a cotton swap which was pre-moistened in aqueous sampling buffer containing disodium phosphate (12.49 g/L, Merck), potassium dihydrogen phosphate (0.63 g/L, Merck) and 1% Triton X-100 (Sigma). The swap was vigorously shaken in a tube containing 2 mL of sampling buffer and then removed. The sample was stored at −20 °C before DNA extraction. Skin hydration and sebum content were measured with a Corneometer (Courage + Khazaka electronic) and Sebumeter (Courage + Khazaka electronic), respectively. None of the volunteers had a history of skin disease, nor had undergone treatment with topical medicine or antibiotics in the last 6 months. The volunteers were recruited in Germany. Written informed consent was obtained from all volunteers and the study was approved by International Medical & Dental Ethics Commission GmbH (IMDEC), Freiburg (Study no. 67885).

**Cultivation of swab sample, CFU count and species identification**. The swab samples were diluted (back, cheek, forehead skin sample: 1:10 and 1:1000; forehead

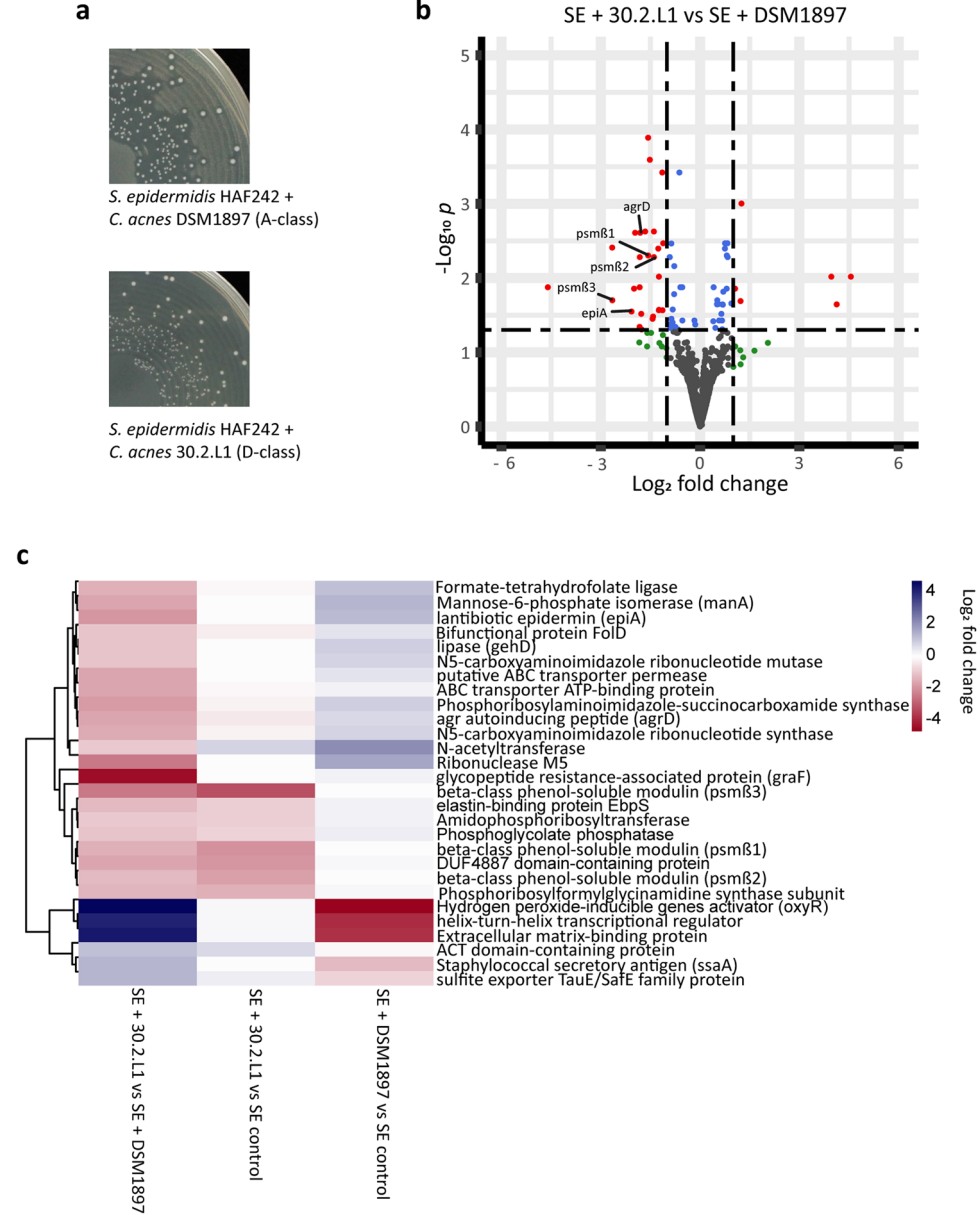

**Fig. 6 Differential expression of quorum-sensing regulated genes in *S. epidermidis* HAF242 co-cultured with *C. acnes* 30.2.L1 (D-class) and *C. acnes* DSM1897 (A-class). a** Colonies of *S. epidermidis* HAF242 exhibit inhibition zones on a lawn of *C. acnes* DSM1897 (A-class) but not on a lawn of *C. acnes* 30.2.L1 (D-class). **b** Differential gene expression of *S. epidermidis* HAF242 (SE) grown in the two co-cultures (co-culture with *C. acnes* 30.2.L1 (D-class) versus co-culture with *C. acnes* DSM1897 (A-class)) (FDR-adjusted *p*-value, cut off: $p \leq 0.05$ and fold-change >2 or <−2). **c** Heat map of all differentially expressed genes in *S. epidermidis* HAF242 when grown in the two co-cultures (first column; genes encoding hypothetical proteins were excluded). Gene expression was also compared between *S. epidermidis* grown in co-culture (second and third columns) versus *S. epidermidis* grown in monoculture ("SE control").

skin sample: 1:1 and 1:100) in 0.9% NaCl solution. Cultivation was done by plating on Columbia agar with 5% sheep blood; agar plates were incubated at 37 °C for 24 h. CFU count was determined with an automatic colony counter (IUL). Up to five colonies that resembled staphylococci based on colony size and colour were randomly picked of each plate and pure cultures were obtained by sub-cultivation on the same agar. Each isolate (572 isolates in total) was assigned to species level by MALDI-TOF mass spectrometry (Supplementary Data 7).

**DNA extraction from skin swab samples**. Prior to DNA extraction, skin swab samples were centrifuged (8.000 × *g*, 30 min at 4 °C), and the supernatant was discarded. The pellets were lysed by using lysostaphin (0.05 mg/mL, Sigma) and lysozyme (9.5 mg/mL, Sigma). DNA was extracted using the DNeasy PowerSoil Kit (QIAGEN), following the manufacturer's instructions. DNA concentrations were measured with the Qubit dsDNA HS Assay (ThermoFisher Scientific) using a Qubit fluorometer.

**Amplicon polymerase chain reaction (PCR)**. The tuf2 amplicon PCR (for staphylococcal population analysis) was performed as described previously[30] using the primers tuf2_fw, 5′-ACAGGCCGTGTTGAACGTG-3′ and tuf2_rev, 5′-ACAG-TACGTCCACCTTCACG-3′. The SLST amplicon fragment (for *C. acnes* population analysis) was amplified using the primers: 5′-TTGCTCGCAACTGCAAGCA-3′ and 5′-CCGGCTGGCAAATGAGGCAT-3′. PCR reaction mixtures were made in a total volume of 25 µl and comprised 5 µl of DNA sample, 2.5 µl AccuPrime PCR Buffer II (Invitrogen, Waltham, MA, USA), 1.5 µl of each primer (10 µM) (DNA Technology, Risskov, Denmark), 0.15 µl AccuPrime Taq DNA Polymerase High Fidelity (Invitrogen, Waltham, MA, USA), and 14.35 µl of PCR grade water. The PCR reaction was performed using the following cycle conditions: initial denaturation at 94 °C for 2 min, 35 cycles of denaturation at 94 °C for 20 s, annealing at 55 °C for 30 s, elongation at 68 °C for 1 min, final elongation step at 72 °C for 5 min. PCR products were verified on an agarose gel and purified using the Qiagen Generead™ Size Selection kit (Qiagen, Hilden, Germany). The concentration of the purified PCR products was measured with a NanoDrop 2000 spectrophotometer (ThermoFisher Scientific, Waltham, MA, USA).

**Amplicon-based next-generation sequencing**. PCR products were used to attach indices and Illumina sequencing adaptors using the Nextera XT Index kit (Illumina, San Diego). Index PCR was performed using 5 µl of template PCR product, 2.5 µl of each index primer, 12.5 µl of 2x KAPA HiFi HotStart ReadyMix and 2.5 µl PCR grade water. Thermal cycling scheme was as follows: 95 °C for 3 min, 8 cycles of 30 s at 95 °C, 30 s at 55 °C and 30 s at 72 °C and a final extension at 72 °C for 5 min. Quantification of the products was performed using the Quant-iT dsDNA HS assay kit and a Qubit fluorometer (Invitrogen GmbH, Karlsruhe, Germany) following the manufacturer's instructions. MagSi-NGS$^{PREP}$ Plus Magnetic beads (Steinbrenner Laborsysteme GmbH, Wiesenbach, Germany) were used for purification of the indexed products as recommended by the manufacturer and normalization was performed using the Janus Automated Workstation from Perkin Elmer (Perkin Elmer, Waltham Massachusetts, USA). Sequencing was conducted using Illumina MiSeq platform using dual indexing and MiSeq reagent kit v3 (600 cycles) as recommended by the manufacturer.

**Amplicon-based NGS sequence data analysis and visualization**. FASTQ sequences obtained after demultiplexing the reads and trimming the primers were imported into QIIME2 (v. 2019.7)[55]. Sequences with an average quality score lower than 20 or containing unresolved nucleotides were removed from the dataset with the split_libraries_fastq.py script from QIIME. The paired-end reads were denoised and chimeras removed with DADA2 via QIIME2, and a feature table was generated[56]. These features were then clustered with VSEARCH at a cut-off of 99% identity against allele databases. The database for the staphylococcal amplicon scheme contained all *tuf* alleles from all staphylococcal genomes available in GenBank (as of December 2019). The database for the *C. acnes* SLST amplicon scheme is available online (http://medbac.dk/slst/pacnes/protocol).

Low abundant features were filtered with a threshold of 2.5%, and figures were prepared in R (v. 4.0.1) with the packages phyloseq[57], ggplot2[58] and gplots[59]. Shannon index was calculated using operational taxonomic unit (OTU) reads.

**Whole genome sequencing of S. epidermidis isolates**. *S. epidermidis* isolates (*n* = 69) were randomly selected for genome sequencing and cultivated on Columbia agar with 5% sheep blood for 24 h at 37 °C. Bacteria were lysed with lysostaphin (0.05 mg/mL, Sigma) and genomic DNA was extracted using the DNeasy UltraClean Microbial Kit by following manufacturer's instructions. DNA concentration and purity were measured by Nanodrop. DNA integrity was checked with Genomic DNA ScreenTape (Agilent) at the 4200 TapeStation System. Sequencing was done as described previously[31]. GenBank accession numbers of the sequenced genomes are listed in Supplementary Data 2.

**Phylogenetic and pan-genomic analysis of S. epidermidis and C. acnes**. Genomes of *S. epidermidis* isolates (*n* = 69) of this study and genomes off *S. epidermidis* (*n* = 286) with N50 >100 kb, taken from the NCBI RefSeq database (status 03.04.2020) were aligned and clustered based on SNVs in their core genome using Parsnp (v 1.0), with enabled recombination filter[60]. Their ST-type was determined with CGE Bacterial Analysis Pipeline using the tool MLST (v 1.6)[61]. Visualization of the tree was done with iTOL (v 5.7)[62]. Information regarding accession numbers of all relevant genomes are contained in Supplementary Data 2 and 8. The presence/absence of the genes *icaA* (query locus tag: SEU43366), *mecA* (AHA36637) and IS256 (D9V02_13220) were determined by blastn. For pan-genomic analyses (69 *S. epidermidis* isolates of this study and 75 *C. acnes* genomes taken from the GenBank database) the Anvi'o[63] tool was used, following Anvi'o workflow for microbial pangenomics (https://merenlab.org/2016/11/08/pangenomics-v2/). Information regarding accession numbers of all relevant genomes are contained in Supplementary Data 2 and 9.

**Antagonistic plate assay**. All 572 CoNS isolates were screened for antimicrobial properties against the indicator strains *S. aureus* DSM799 and *C. acnes* DSM1897. Therefore, bacterial lawn plates were prepared. Liquid cultures of *C. acnes* indicator strains and *S. aureus* were prepared in CASO broth. For staphylococci the liquid culture was adjusted to an optical density of $OD_{600nm} = 0.002$; while for *C. acnes* strain cultures were adjusted to $OD600 = 0.075$. 6 mL of the adjusted culture was pipetted onto a rectangular Tryptic Soy Agar (TSA) plate and distributed evenly. For round TSA plates 3 mL bacterial suspension was used. After 30 s excess liquid was removed and the plates were dried for 4 h. The plates were stored up to 3 weeks at 4 °C.

The CoNS isolates were cultivated for 20 h at 37 °C shaking in 1 mL CASO broth in 96-Deepwell plates. The 96-Deepwell plate were centrifuged at 2000 rpm for 5 min, 500 µL supernatant was removed and the pellet was re-suspended in the remaining liquid. The concentrated bacterial cultures were transferred into 96-well U-bottom plates. With a replicator stamps bacterial cultures were transferred on rectangular lawn plates. After 4 h of drying, the plates were cultivated with varying conditions (*S. aureus* lawn plates: 24 h, 37 °C; *C. acnes* lawn plates: 4–5 days, 37 °C, in anaerobic container with AnaeroGen bag (Thermo Scientific)). A visible inhibition zone around a staphylococcal colony was regarded as antimicrobial activity. Staphylococcal strains that showed antimicrobial properties were verified in triplicates. These strains were further tested against eleven different *C. acnes*

indicator strains from six different SLST classes. Strain names and accession numbers of all indicator strains are listed in Supplementary Table 4.

**Co-cultures of S. epidermidis and C. acnes**. Lawn plates with *C. acnes* DSM1897 and *C. acnes* 30.2.L1 were prepared as described above. A liquid culture of *S. epidermidis* HAF242 was grown to exponential growth phase and diluted 1:10$^6$ in 0.9% NaCl solution and plated on the *C. acnes* lawn plates with a spiral plater (Don Whitley Scientific). Plates were incubated for 4 h at 37 °C under aerobic conditions and then 72 h in anaerobic conditions (AnaeroGen bag (Thermo Scientific)) at 37 °C. The bacteria were harvested using a cell spreader and suspended in 10 mL 0.9% NaCl solution and immediately frozen at −80 °C. Experiments were done in triplicates.

**RNA extraction and RNA sequencing**. Harvested cells were resuspended in 800 µl RLT buffer (RNeasy Mini Kit, Qiagen) with β-mercaptoethanol (10 µl/ml) and cell lysis was performed using a laboratory ball mill. Subsequently, 400 µl buffer RLT (RNeasy Mini Kit Qiagen) with β-mercaptoethanol (10 µl/ml) and 1200 µl 96% [v/v] ethanol were added. For RNA isolation, the RNeasy Mini Kit (Qiagen) was used as recommended by the manufacturer, but instead of buffer RW1, the buffer RWT (Qiagen) was used in order to also isolate RNAs smaller 200 nt. To determine the RNA integrity number (RIN) the isolated RNA was run on an Agilent Bioanalyzer 2100 using an Agilent RNA 6000 Nano Kit, as recommended by the manufacturer (Agilent Technologies, Waldbronn, Germany). Remaining genomic DNA was removed by digestion with TURBO DNase (Invitrogen, ThermoFischer Scientific, Paisley, UK). The Illumina Ribo-Zero plus rRNA Depletion Kit ((Illumina Inc., San Diego, CA, USA) was used to reduce the amount of rRNA-derived sequences. For sequencing, strand-specific cDNA libraries were constructed with a NEBNext Ultra II directional RNA library preparation kit for Illumina and the NEBNext Multiplex Oligos for Illumina (New England BioLabs, Frankfurt am Main, Germany). To assess quality and size of the libraries, samples were run on an Agilent Bioanalyzer 2100 using an Agilent High Sensitivity DNA Kit, as recommended by the manufacturer (Agilent Technologies, Waldbronn, Germany). Concentration of the libraries were determined using the Qubit® dsDNA HS Assay Kit, as recommended by the manufacturer (Life Technologies GmbH, Darmstadt, Germany). Sequencing was performed on a NovaSeq 6000 instrument (Illumina Inc., San Diego, CA, USA) using NovaSeq 6000 SP Reagent Kit v1.5 (100 cycles) and the NovaSeq XP 2-Lane Kit v1.5 for sequencing in the paired-end mode and running 2 × 50 cycles. For quality filtering and removing of remaining adaptor sequences, Trimmomatic-0.39[64] and a cutoff phred-33 score of 15 were used. Mapping against the reference genome was performed with Salmon (v 1.5.2)[65]. As mapping backbone a file that contained all annotated transcripts excluding rRNA genes and the whole genome sequence of the reference as decoy was prepared with a k-mer size of 11. Decoy-aware mapping was done in selective-alignment mode with "–mimicBT2", "–disableChainingHeuristic", and "–recoverOrphans" flags as well as sequence and position bias correction. For –fldMean and –fldSD, a value of 325 and 25 was used, respectively. The quant.sf files produced by Salmon were subsequently loaded into R (v 4.0.3) using the tximport package (v 1.18.0)[66]. DeSeq2 (v 1.30.0)[67] was used for normalization of the reads; foldchange-shrinkages were also calculated with DeSeq2 and the apeglm package (v 1.12.0)[68]. Genes with a log$_2$-fold change of +2/ −2 and a *p*-adjust value <0.05 were considered differentially expressed.

**Statistics and reproducibility**. Statistical analysis was done in R (v. 4.0.1) using the packages ggplot2 (v. 3.3.5), phyloseq (v. 1.34.0), gplots (v 3.1.1.), corrplot (v 0.90)[69], ANCOMBC (v 1.0.5)[70] pheatmap (v 1.0.12)[71] and EnhancedVolcano (v 1.8.0)[72]. Unpaired two-sided Wilcoxon was used for comparison of two groups. The samples sizes are given in the main text or figure legends. Correlation analysis was done with Spearman analysis and visualized with ggplot2 and corrplots. Differential abundance between was calculated with the ANCOMBC package (100 max. iterations, 0.80 zero cut-off). In case of multiple comparisons *p* values were FDR-adjusted with the Holm method. Three independent biological replicates were performed for the antagonistic assay to verify all staphylococcal strains that showed antimicrobial properties. Co-culture experiments for transcriptome sequencing were done in three independent biological replicates.

**Reporting summary**. Further information on research design is available in the Nature Research Reporting Summary linked to this article.

## Data availability

Whole genome sequence data (69 *S. epidermidis* genomes) generated for this study are deposited in GenBank with the bioproject number PRJNA793831 and can be accessed here. The closed whole genome sequence of *S. epidermidis* HAF242 is deposited in GenBank with the accession numbers CP090941 (chromosome) and CP090942-CP090944 (plasmids). The amplicon-based NGS data is stored at SRA with the bioproject number PRJNA795320 and can be accessed here. The transcriptome sequencing data (*S. epidermidis* HAF242 in mono- and co-cultures) is stored at SRA with the bioproject number PRJNA801462 and can be accessed here. All other data are available from the

corresponding authors on reasonable request. All data sets used for figures and the GenBank accession numbers of all bacterial genomes used in this study are provided in the supplementary data (Supplementary Data 1–10).

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

## Acknowledgements

This work was supported by grants from the NovoNordisk Foundation (grant no. NNF18OC0053172) and the Leo Foundation (LF-OC-21-000826) to H.B. We thank Lise Hald Schultz for excellent technical assistance, and Lesley Ann Ogilvie for carefully reading the manuscript. Data processing was performed on the GenomeDK cluster; we would like to thank GenomeDK and Aarhus University for providing computational resources.

## Author contributions

C.M.A., W.R.S., H.W., J.H.R., J.H. and H.B. contributed to the conception and design of the study. C.M.A. performed wet lab benchwork and analysed data. A.P. and M.B. contributed to sequence data generation and C.M.A., K.S.J., A.P. and H.B. analysed sequence data. C.M.A. and H.B. wrote the manuscript and all authors contributed to manuscript revision and read and approved the final version.

## Competing interests
The authors declare the following competing interests: C.M.A., H.W., J.H.R. and J.H. are employees at Beiersdorf AG. The other authors declare no competing interests.

## Ethics statement
The study was approved by International Medical & Dental Ethics Commission GmbH (IMDEC), Freiburg, Germany (Study no. 67885). Written informed consent was obtained from all volunteers.
