## [Peer Review File · Communications Biology]

Reviewers' comments:

Reviewer #1 (Remarks to the Author):

Ahle et al. Interference and co-existence of staphylococci and cutibacteria ...

Here, the distribution of staphylococci and cutibacteria at 4 different human skin sites of 30 healthy individuals was investigated at phylotype resolution using amplicon-based next-generation sequencing.

572 bacterial isolates were obtained via 86 selective cultivations, of which 557 were identified as CoNS via MALDI-TOF mass spectrometry. Across all skin sites, the majority of isolates were identified as *S. epidermidis* (67.2 %), followed by *Staphylococcus hominis* (15.4 %). Forehead, cheek and back skin sites were 89 dominated by strains of *S. epidermidis*, followed by *Staphylococcus capitis*, whereas on forearm skin sites, a larger number of strains of *S. hominis* and *Staphylococcus haemolyticus* were isolated. *S. epidermidis* strains from healthy human skin are highly diverse and belong to non-nosocomial associated phylogenetic lineages.

On the human skin 39 different *C. acnes* SLST types were identified belonging to 10 SLST classes (A to L). *C. acnes* strains belonging to the IA1 phylotype were most frequently detected, followed by phylotype II.

By investigating the antimicrobial activity against *C. acnes* they found that activity against acne- but not healthy skin associated *C. acnes* phlotypes. To better understand the selective bioactivity CoNS to different *C. acnes* strains a transcriptome analysis with co-cultures was carried out. Two model bacteria have been chosen, *S. epidermidis* HAF242 and A-class *C. acnes* DSM1897, the latter is sensitive to the staphylococcal bacteriocin, which was epidermin. The other co-cultures was *S. epidermidis* HAF242 and *C. acnes* 30.2.L1, the latter was resistant to epidermin. *C. acnes* 30.2.L1 triggered in *S. epidermidis* HAF242 downregulation of *agrD*, *psms*' and *epiA*. Particularly the downregulation of *epiA* could explain that *C. acnes* 30.2.L1 is tolerant to epidermin. This indeed suggests that there is a selective microbial interference that contributes to healthy skin homeostasis.

All in all, this is a very solid and comprehensive work that provides new insights into the colonization of the skin with the dominant skin bacteria such as staphylococci and cutibacteria. Especially the classification into phylogroups and the selective toleration of non-acne phlotypes by bacteriocin-producing CoNS is interesting. The topic fits perfectly into the subject area of communications biology.

There are few points of criticism:

Line 51ff:

In the introduction one could quote two papers on skin microbiota and microbiom analysis that show that *S. epidermidis* and particularly trace amines (TA)-producing *Staphylococcus epidermidis* strains expressing *SadA* are predominant on human skin:

Luqman et al. (2020a) Trace amines produced by skin bacteria accelerate wound healing in mice. *Commun Biol* 3: 277.

Luqman et al. (2020b) The Neuromodulator-Encoding *sadA* Gene Is Widely Distributed in the Human Skin Microbiome. *Front Microbiol* 11: 573679.

It should be made clear which of the *C. acnes* classes are acne producers and which not. This should be mentioned in Table 1 but should be also made clearer in the text. It should be also mentioned by which markers (physiological, genetic) the acne producing classes differ from the healthy ones.

The Fig 2e, Fig. 4d and Fig 5a are difficult to understand. Maybe a Table would be clearer.

Reviewer #2 (Remarks to the Author):

The study investigate the distribution of common human skin bacteria, namely staphylococci and cutibacterium acnes, on four selected skin sites of healthy individuals. The study is based on NGS methods: targeted sequencing of the *tuf* gene for staphylococcal diversity analysis, targeted amplicon sequencing of a *C. acnes* specific gene for analysis of *C. acnes* clonal diversity and WGS of *S. epidermidis* to analyse the clonal diversity on the skin. Furthermore, the authors have performed co-culturing assays in order to investigate if CoNS strains isolated from healthy skin is able to antagonize *C. acnes* strains associated with acne, and find that 30/557 CoNS strains were able to inhibit *C. acnes*. The authors also performed transcriptomics to investigate *S. epidermis* gene expression during co-culturing with *C. acnes* strains. I believe that this part of the manuscript is the most interesting and original part.

The methods used is all highly suitable for the aim of this study, and I believe that the results shown gives new insight into the role of inter-genera bacterial competition within the human skin. The manuscript is well written. I have some minor questions and comments.

Comments and questions:

Line32: [SEP] You write that you profiled the staphylococcal and *C. acnes* landscapes at phylotype resolution. However, the phylotype resolution was only performed for *S. epidermidis* and not other staphylococcal species . Please change the wording in the abstract.

Fig2: [SEP] In the figure text it says 120 samples from 30 individuals. However, there is not 4x30 bars in the barplot in fig 2.a (20 back samples, 26 cheek samples, 21 forearm samples, and 26 forehead samples as far as I can see). Why are some samples not included in the bar charts? Was they excluded during QC filtering? Please explain.

It is difficult to discriminate between the red colors used to highlight *S. capitis* and *S. lugdunensis*. Is it possible to change one of the colors?

Chapter starting at line136:

You write that none of the *S. epi* strains belonged to the hospital associated lineages ST2, ST5 and ST23. However, you don't mentioned which STs that was the most commonly observed on the four skin sites. Would be interesting to get this information, and not only information of clades. Also, did you find ST215, which has been associated with PJI in Sweden? [SEP] With a total of 69 *S. epi* isolates I assume that you have isolates collected from more than one skin site for some of the individuals. Was it the same STs you find across skin sites within individuals?

S. epi is polyclonal and thus >1 ST types often co-colonizes the human skin (fx shown in Rendboe 2020, BMC microbiology). Did you collect multiple strains from the same samples in order to look at the diversity within a single site? If not, you might want to mention this as a limitation in the Discussion section, as you can't rule out that some of the participants in fact carried a hospitals associated ST type (though it might be the most abundant within the the samplingsite).

Line 266-269:

This is very interesting! Did you compare the accessory genomes of the sensitive and resistant *C.acnes* strain in order to look for genes within the 30.2.L1 strain that might explain this effect? Or performed transcriptomics to compare the gene expression within the two *C. acnes* strains?

Line393: What was the country of origin of the participants? Germany? Please write as it is informative to know this when comparing between studies.

Line 399: You write that the samples was kept at -20C. It has been shown that bacteria is able to grow in transport media at this temperature. For how long was the samples stored at -20 before

processing? And did you perform any pilot study, testing if bacterial densities did not increase within in the sampling buffer used within the stored time period?

Line 462: Should be OTU instead of OUT (minor misspelling)

Line 473: Did you look for (and perhaps) removed any recombinations within the *S. epidermidis* genomes before constructing the phylogeny?

Reviewer #3 (Remarks to the Author):

Ahle and colleagues report the characterization of interactions between *Staphylococcus* species and *Cutibacterium acnes* in healthy skin. Molecular interactions between skin commensals is a research area close to my heart and the study by Ahle contributes to our understanding of how two important genera of bacterial residents interact and co-exist with each other. I especially appreciated that they not only infer interactions (co-stimulatory or co-inhibitory) from their sequencing data, but also test these using in vitro co-cultivation assays. In an alternative approach to profiling entire microbiome communities, the authors performed targeted high throughput sequencing to zoom in on characterization of the different lineages of *staphylococcus* and *C. acnes* respectively, which is the main focus of the manuscript. They identified and isolated several novel *S. epidermidis* healthy skin isolates and several of these were selected for full genome sequencing. These isolates spanned the spectrum of previously identified lineages, yet none of these belonged to a pathogenicity-related sequence-type. It is encouraging for skin microbiota research that the authors isolation of *Staph* spp. compares to the overall microbiota composition in these body sites, and that there is no major over/under sampling of specific species. In a large co-culture experiment, the authors identified 30 out of 557 *staph* strains which produced an antimicrobial activity against *C. acnes*. Some of these were specific against clades of *C. acnes* that are associated with acneic skin. I enjoyed reading the manuscript, I believe the experimental setup and execution is solid and the data mostly support the claims. During my review, I only identified some minor points which might improve the manuscript, and listed these below for the authors to consider.

Jan Claesen

Line 35, line 235: 'selective deregulation of the antimicrobial activity' Not really familiar with this term, perhaps reword?

Fig. 1b: This shows the #of isolates of *Staphylococcus* species from the different skin sites, but the legend also includes Gram-pos cocci, *M. luteus* or even *Candida*. I would only show the *Staph* species as in the legend heading or correct the heading.

Fig. 4 and 5, Spearman correlations only shown for the top 4 *C. acnes* SLSTs. Perhaps expand the table to include all SLSTs (in main figure or SI)?

Related to this (line 197 and below): *S. epidermidis* as a species is still quite variable (as the authors point out in their analysis of the core vs. accessory genomes above), is it possible to increase the resolution, analyzing correlations between different *S. epidermidis* STs (for example using the characterization of the cultivated isolates) and the *C. acnes* SLSTs? At this point, it is unclear to me why this correlation inferred from the sequencing matters, since all 572 CoN-*Staph* isolates were screened for bioactivity against *C. acnes* and only five of the 30 isolates that produce activity were *S. epi*.

Lines 211-214, I might include brief descriptors (healthy-skin associated/acneic skin-associated) in the narrative on A-, D- and H-class *C. acnes* strains to remind/help guide the non-specialist reader of the characteristics of these classes.

Line 219: 'in skin sites that contained a staphylococcal strain exhibiting antimicrobial activity'. This is an inference as the antimicrobial activity has not been demonstrated for these isolates from the skin sites. Better wording would be along the lines of 'in skin sites that contained a staphylococcal strain related to 'X-lineage' type which we showed displayed antimicrobial activity.' Unless you identify genes responsible for the anti-staph activity and track the presence of these in the microbiome samples, it is hard to justify the claim that they exhibit antimicrobial activity solely based on clade membership. I agree this could be due to epidermin or PSMs, but this has not been verified by mass spec.

Just out of curiosity. Epidermin is a well-characterized lantibiotic produced by *S. epi* strains. Is the cluster from HAF242 predicted to produce the same compound as the original type strain *S. epidermidis* Tü 3298 (i.e. are their EpiA structural peptides identical?) In addition, was the epidermin cluster identified in any of the other sequenced isolates from this study (that perhaps did not produce antibiotic activity)? Did the authors investigate whether the activity is due to epidermin production (for example by detection with mass spectrometry)? Also, the significant downregulation of the phenol-soluble modulins in the transcriptomic experiment that compares co-cultured versus monoculture (and the relatively unchanged *epiA* expression), taken together with the huge changes in *oxyR* (which is not discussed), to me suggests that the (reactive oxygen species-inducing capacity of the) PSMs might be responsible for the observed antimicrobial activity against the susceptible *C. acnes* strain (rather than epidermin)? I think this is an interesting and novel observation regarding Staph-*C. acnes* interspecies activities.

Line 460: I don't quite understand how the data normalization was performed. Separate amplifications were used for identification of Staph and *C. acnes* relative abundances can be compared across samples. Is there a control (qPCR) or internal standard that is used for normalization or is there no cross-comparison of *S. epi* and *C. acnes* there are in each skin site?

Reviewer #1 (Remarks to the Author):

Ahle et al. Interference and co-existence of staphylococci and cutibacteria ...

Here, the distribution of staphylococci and cutibacteria at 4 different human skin sites of 30 healthy individuals was investigated at phylotype resolution using amplicon-based next-generation sequencing.

572 bacterial isolates were obtained via 86 selective cultivations, of which 557 were identified as CoNS via MALDI-TOF mass spectrometry. Across all skin sites, the majority of isolates were identified as *S. epidermidis* (67.2 %), followed by *Staphylococcus hominis* (15.4 %). Forehead, cheek and back skin sites were dominated by strains of *S. epidermidis*, followed by *Staphylococcus capitis*, whereas on forearm skin sites, a larger number of strains of *S. hominis* and *Staphylococcus haemolyticus* were isolated. *S. epidermidis* strains from healthy human skin are highly diverse and belong to non-nosocomial associated phylogenetic lineages.

On the human skin 39 different *C. acnes* SLST types were identified belonging to 10 SLST classes (A to L). *C. acnes* strains belonging to the IA1 phylotype were most frequently detected, followed by phylotype II.

By investigating the antimicrobial activity against *C. acnes* they found that activity against acne- but not healthy skin associated *C. acnes* phlotypes. To better understand the selective bioactivity CoNS to different *C. acnes* strains a transcriptome analysis with co-cultures was carried out. Two model bacteria have been chosen, *S. epidermidis* HAF242 and A-class *C. acnes* DSM1897, the latter is sensitive to the staphylococcal bacteriocin, which was epidermin. The other co-cultures was *S. epidermidis* HAF242 and *C. acnes* 30.2.L1, the latter was resistant to epidermin. *C. acnes* 30.2.L1 triggered in *S. epidermidis* HAF242 downregulation of *agrD*, *psms'* and *epiA*. Particularly the downregulation of *epiA* could explain that *C. acnes* 30.2.L1 is tolerant to epidermin. This indeed suggests that there is a selective microbial interference that contributes to healthy skin homeostasis.

All in all, this is a very solid and comprehensive work that provides new insights into the colonization of the skin with the dominant skin bacteria such as staphylococci and cutibacteria. Especially the classification into phylogroups and the selective toleration of non-acne phlotypes by bacteriocin-producing CoNS is interesting. The topic fits perfectly into the subject area of communications biology.

There are few points of criticism:

Line 51ff:

In the introduction one could quote two papers on skin microbiota and microbiom analysis that show that *S. epidermidis* and particularly trace amines (TA)-producing *Staphylococcus epidermidis* strains expressing *SadA* are predominant on human skin:

Luqman et al. (2020a) Trace amines produced by skin bacteria accelerate wound healing in mice. *Commun Biol* 3: 277.

Luqman et al. (2020b) The Neuromodulator-Encoding *sadA* Gene Is Widely Distributed in the Human Skin Microbiome. *Front Microbiol* 11: 573679.

We have added these references in the introduction (line 49) as references 11 and 12.

It should be made clear which of the *C. acnes* classes are acne producers and which not. This should be mentioned in Table 1 but should be also made clearer in the text. It should be also mentioned by which markers (physiological, genetic) the acne producing classes differ from the healthy ones.

The notion which *C. acnes* SLST classes are enriched on acne-affected skin is written in the introduction in lines 58-63. We added this information also in the table 1 and in the results (lines 225/226). In the discussion we elaborated on this in lines 345 to 362 (with four added references #38-41).

Regarding the markers of acne-associated strains we would like to refer to recent studies and reviews that have highlighted numerous possible genetic and physiological differences between acne-associated and healthy skin associated strains (see lines 352-356); one recently discussed distinguishing element might be porphyrin production (see lines 354-356). However, we think that many aspects are not fully understood and would like to point out that the delineation of acne- and healthy skin-associated SLST classes might be oversimplified, as it seems likely that a high diversity of strains belonging to different SLST classes forms the basis of a healthy skin microbiome and thus, the loss of diversity is associated with acne.

The Fig 2e, Fig. 4d and Fig 5a are difficult to understand. Maybe a Table would be clearer.

We extended the figure legends of Fig 2e, 4d, 5a, to better explain how to read the Spearman correlation plots and added an example how to read it in each figure legend.

Reviewer #2 (Remarks to the Author):

The study investigate the distribution of common human skin bacteria, namely staphylococci and cutibacterium acnes, on four selected skin sites of healthy individuals. The study is based on NGS methods: targeted sequencing of the *tuf* gene for staphylococcal diversity analysis, targeted amplicon sequencing of a *C. acnes* specific gene for analysis of *C. acne* clonal diversity and WGS of *S. epidermidis* to analyse the clonal diversity on the skin. Furthermore, the authors have performed co-culturing assays in order to investigate if CoNS strains isolated from healthy skin is able to antagonize *C. acne* strains associated with acne, and find that 30/557 CoNS strains were able to inhibit *C. acne*. The authors also performed transcriptomics to investigate *S. epidermidis* gene expression during co-culturing with *C. acne* strains. I believe that this part of the manuscript is the most interesting and original part.

The methods used is all highly suitable for the aim of this study, and I believe that the results shown gives new insight into the role of inter-genera bacterial competition within the human skin. The manuscript is well written. I have some minor questions and comments.

Comments and questions:

Line32: You write that you profiled the staphylococcal and *C. acnes* landscapes at phylotype resolution. However, the phylotype resolution was only performed for *S. epidermidis* and not other staphylococcal species. Please change the wording in the abstract.

We changed the abstract accordingly

Fig2: In the figure text it says 120 samples from 30 individuals. However, there is not 4x30 bars in the barplot in fig 2.a (20 back samples, 26 cheek samples, 21 forearm samples, and 26 forehead samples as far as I can see). Why are some samples not included in the bar charts? Was they excluded during QC filtering? Please explain.

Thank you for pointing this out. The PCR reaction of the two amplicons did not work for all 120 samples, possibly due to very low DNA concentrations in some samples or the presence of PCR-inhibiting substances in a few samples. For the *C. acnes* SLST fragment amplification, 7 out of 120 samples failed, and for the *tuf2* amplification 27 out of 120 samples failed. We added this information in the results part (pages 6 and 12) and in the respective figure legends (Fig. 2 and Fig. 4)

It is difficult to discriminate between the red colors used to highlight *S. capitis* and *S. lugdunensis*. Is it possible to change one of the colors?

We changed the color code (*S. lugdunensis* is now depicted in a brown color). Moreover, we added the data now also in excel format (Supplementary Data 1).

Chapter starting at line136:

You write that none of the *S. epi* strains belonged to the hospital associated lineages ST2, ST5 and ST23. However, you don't mentioned which STs that was the most commonly observed on the four skin sites. Would be interesting to get this information, and not only information of clades. Also, did you find ST215, which has been associated with PJI in Sweden? With a total of 69 *S. epi* isolates I assume that you have isolates collected from more than one skin site for some of the individuals. Was it the same STs you find across skin sites within individuals?

Thank you for mentioning this. We added additional information on page 10 and in the supplement (Supplementary data 2; this table now includes information about the STs of the 69 *S. epidermidis* isolates, and their origin (volunteer, skin site)). Most common were isolates of ST19 (9 isolates from 5 people), ST73 (9 isolates from 7 people) and ST65 (8 isolates from 6 people). Isolates of ST215 were not found (information added on page 10; reference added regarding ST215 (reference 17)). From 19 individuals we have isolated and sequenced more than one *S. epidermidis* isolate (two to six isolates per person). Almost all individuals (18/19) carried multiple *S. epidermidis* strains belonging to different STs. For example, one person (volunteer 13) from which we isolated six *S. epidermidis* strains: these belonged to five different STs. Only from one

volunteer (volunteer 3) the same ST was found for all four sequenced *S. epidermidis* strains (that were isolated from four different skin sites).

S. epi is polyclonal and thus >1 ST types often co-colonizes the human skin (fx shown in Rendboe 2020, BMC microbiology). Did you collect multiple strains from the same samples in order to look at the diversity within a single site? If not, you might want to mention this as a limitation in the Discussion section, as you can't rule out that some of the participants in fact carried a hospital-associated ST type (though it might be the most abundant within the sampling site).

It is true that we do not have a complete picture about the *S. epidermidis* population, in particular in single samples regarding the presence of multiple STs. We added this as a limitation in the discussion section (page 26). Only from three samples (volunteer 2 forehead; volunteer 13 forehead; volunteer 24 forehead) multiple *S. epidermidis* strains were sequenced; all three have in the single skin site *S. epidermidis* strains belonging to different STs (supplementary data 2), underlining the findings of Rendboe et al. (the reference was added as reference 54).

Line 266-269:

This is very interesting! Did you compare the accessory genomes of the sensitive and resistant *C. acnes* strain in order to look for genes within the 30.2.L1 strain that might explain this effect? Or performed transcriptomics to compare the gene expression within the two *C. acnes* strains?

We are planning follow-up research to investigate these findings further, including comparative genomics and comparative transcriptomics, with a focus on *C. acnes*.

Line 393: What was the country of origin of the participants? Germany? Please write as it is informative to know this when comparing between studies.

The study was performed in Germany. We added this information in the methods section.

Line 399: You write that the samples were kept at -20°C. It has been shown that bacteria is able to grow in transport media at this temperature. For how long were the samples stored at -20 before processing? And did you perform any pilot study, testing if bacterial densities did not increase within the sampling buffer used within the stored time period?

We did not use "classical" transport medium. The sampling buffer we used contained no carbon source. We did not detect any growth of bacteria in this sampling buffer. We rather see a reduction of CFU after long-term storage. Samples were stored at -20 degrees for a maximum of 5 weeks before DNA isolation.

Line 462: Should be OTU instead of OUT (minor misspelling)

This was corrected.

Line 473: Did you look for (and perhaps) removed any recombinations within the *S. epidermidis* genomes

before constructing the phylogeny?

We used the filter for recombination detection in parsnp (now mentioned in the methods section) for genome-wide phylogenetic analyses.

Reviewer #3 (Remarks to the Author):

Ahle and colleagues report the characterization of interactions between *Staphylococcus* species and *Cutibacterium acnes* in healthy skin. Molecular interactions between skin commensals is a research area close to my heart and the study by Ahle contributes to our understanding of how two important genera of bacterial residents interact and co-exist with each other. I especially appreciated that they not only infer interactions (co-stimulatory or co-inhibitory) from their sequencing data, but also test these using in vitro co-cultivation assays. In an alternative approach to profiling entire microbiome communities, the authors performed targeted high throughput sequencing to zoom in on characterization of the different lineages of *staphylococcus* and *C. acnes* respectively, which is the main focus of the manuscript. They identified and isolated several novel *S. epidermidis* healthy skin isolates and several of these were selected for full genome sequencing. These isolates spanned the spectrum of previously identified lineages, yet none of these belonged to a pathogenicity-related sequence-type. It is encouraging for skin microbiota research that the authors isolation of *Staph* spp. compares to the overall microbiota composition in these body sites, and that there is no major over/under sampling of specific species. In a large co-culture experiment, the authors identified 30 out of 557 *staph* strains which produced an antimicrobial activity against *C. acnes*. Some of these were specific against clades of *C. acnes* that are associated with acneic skin. I enjoyed reading the manuscript, I believe the experimental setup and execution is solid and the data mostly support the claims. During my review, I only identified some minor points which might improve the manuscript, and listed these below for the authors to consider.

Jan Claesen

Line 35, line 235: 'selective deregulation of the antimicrobial activity' Not really familiar with this term, perhaps reword?

We simplified the wording.

Fig. 1b: This shows the #of isolates of *Staphylococcus* species from the different skin sites, but the legend also includes Gram-pos cocci, *M. luteus* or even *Candida*. I would only show the *Staph* species as in the legend heading or correct the heading.

Thank you for pointing this out. We corrected the figure legend accordingly.

Fig. 4 and 5, Spearman correlations only shown for the top 4 *C. acnes* SLSTs. Perhaps expand the table to include all SLSTs (in main figure or SI)?

We looked at all *C. acnes* SLST classes. However, many of the less abundant SLST classes (e.g. classes B, G and L) are only detectable in a limited number of samples (e.g. G-class *C. acnes* was only detected in five samples). This makes correlation analysis with these less abundant *C. acnes* SLST classes not reliable. This is now mentioned in the figure legends of Figs. 4 and 5.

Related to this (line 197 and below): *S. epidermidis* as a species is still quite variable (as the authors point out in their analysis of the core vs. accessory genomes above), is it possible to increase the resolution, analyzing correlations between different *S. epidermidis* STs (for example using the characterization of the cultivated isolates) and the *C. acnes* SLSTs? At this point, it is unclear to me why this correlation inferred from the sequencing matters, since all 572 CoN-Staph isolates were screened for bioactivity against *C. acnes* and only five of the 30 isolates that produce activity were *S. epi*.

Regarding the resolution of the used approach in our study: the *tuf2*-based NGS method applied here does not allow to differentiate STs of *S. epidermidis*. This could be achieved by using a different approach, e.g. the recently established method by Rendboe et al. (Rendboe et al., 2020). We added this limitation of our study in the discussion (page 26).

Please note that we added some information (on page 10) regarding the STs of the genome-sequenced 69 *S. epidermidis* isolates (see also response to reviewer 2). The analysis underlined a high degree of variability among human skin-associated *S. epidermidis* and highlighted that different STs can coexist on the skin of one person (and even on the same skin site), as it was reported before (Rendboe et al., 2020). We did not isolate (and sequence) enough *S. epidermidis* per skin site (and in total) to correlate between *S. epidermidis* STs and *C. acnes* SLSTs (given the fact that most skin sites seem to be co-colonized by multiple *S. epidermidis* STs). A future study is needed to increase the resolution to the ST level of *S. epidermidis* to address the question if certain STs of *S. epidermidis* have a specific correlation/inverse correlation with certain *C. acnes* SLST classes.

Lines 211-214, I might include brief descriptors (healthy-skin associated/acneic skin-associated) in the narrative on A-, D- and H-class *C. acnes* strains to remind/help guide the non-specialist reader of the characteristics of these classes.

Please see the response to reviewer 1. This was now described more thoroughly throughout the manuscript (introduction (lines 58-63); table 1; results (lines 225/226); discussion (lines 345 to 362)).

Line 219: 'in skin sites that contained a staphylococcal strain exhibiting antimicrobial activity'. This is an inference as the antimicrobial activity has not been demonstrated for these isolates from the skin sites. Better wording would be along the lines of 'in skin sites that contained a staphylococcal strain related to 'X-lineage'

type which we showed displayed antimicrobial activity.’ Unless you identify genes responsible for the anti-staph activity and track the presence of these in the microbiome samples, it is hard to justify the claim that they exhibit antimicrobial activity solely based on clade membership. I agree this could be due to epidermin or PSMs, but this has not been verified by mass spec.

We have reworded this sentence to: “in skin sites from which a staphylococcal strain was isolated that exhibited antimicrobial activity in the antagonistic assay” (page 16). We would like to point out that the presence of antimicrobial activity was not predicted based on clade membership; the antimicrobial activity was tested in the antagonistic assay for all isolated 557 staphylococcal strains that were isolated from the 30 volunteers.

Just out of curiosity. Epidermin is a well-characterized lantibiotic produced by *S. epi* strains. Is the cluster from HAF242 predicted to produce the same compound as the original type strain *S. epidermidis* Tü 3298 (i.e. are their EpiA structural peptides identical?) In addition, was the epidermin cluster identified in any of the other sequenced isolates from this study (that perhaps did not produce antibiotic activity)? Did the authors investigate whether the activity is due to epidermin production (for example by detection with mass spectrometry)? Also, the significant downregulation of the phenol-soluble modulins in the transcriptomic experiment that compares co-cultured versus monoculture (and the relatively unchanged *epiA* expression), taken together with the huge changes in *oxyR* (which is not discussed), to me suggests that the (reactive oxygen species-inducing capacity of the) PSMs might be responsible for the observed antimicrobial activity against the susceptible *C. acnes* strain (rather than epidermin)? I think this is an interesting and novel observation regarding Staph-*C. acnes* interspecies activities.

These are some interesting questions.

The epidermin precursor of *S. epidermidis* HAF242 has the same peptide sequence as the epidermin precursor of strain Tü3298. We do not know if also the mature epidermin has the same structure, but it seems very likely as we did not find sequence differences in the epidermin biosynthesis gene cluster between HAF242 and Tü3298.

Furthermore, we checked the genome-sequenced 69 *S. epidermidis* strain cohort for the presence of *epiA*. We found it only in three strains; these strains were also antimicrobially active. This may suggest that epidermin is indeed responsible for the antimicrobial activity detected in strain HAF242, but we did not investigate epidermin production in this strain (e.g. by mass spectrometry).

PSMs of type beta are encoded in the genomes of many *S. epidermidis* strains in the 69-strain cohort, thus also in strains that have not exhibited any antimicrobial activity in the antagonistic assay. This again could indicate that epidermin might be the main reason for the antimicrobial activity of strain HAF242. However, it seems possible that PSMβs could enhance the antimicrobial effect in an epidermin-positive strain. Future work is needed to clarify the exact nature of the antimicrobial action in strain HAF242.

Line 460: I don't quite understand how the data normalization was performed. Separate amplifications were

used for identification of Staph and *C. acnes* relative abundances can be compared across samples. Is there a control (qPCR) or internal standard that is used for normalization or is there no cross-comparison of *S. epi* and *C. acnes* there are in each skin site?

Separate amplifications (and subsequent NGS) were performed for the identification of relative abundances of *C. acnes* SLSTs and staphylococcal species, respectively. We did not perform quantitative analyses (except the CFU count of staphylococci upon cultivation), thus cross-comparison of staphylococci and *C. acnes* was not possible. We mentioned this limitation in the discussion section (lines 408/409) and corrected the relevant sentence in the methods part (page 30).

REVIEWERS' COMMENTS:

Reviewer #1 (Remarks to the Author):

This is a very good work. All critical comments and questions were answered to the fullest satisfaction. The publication is an important contribution to a better understanding of the coexistence and interaction of bacteria in a particular habitat, such as the skin.

Reviewer #2 (Remarks to the Author):

The corresponding authors have given a good response, and answered all of my questions. I am satisfied. I can recommend the paper for publication. It is a solid and interesting piece of work.

Reviewer #3 (Remarks to the Author):

The authors addressed all my comments and I have nothing further to add. I think this is a great paper as-is.